# Comprehensive Genome-Wide Analysis of Histone Acetylation Genes in Roses and Expression Analyses in Response to Heat Stress

**DOI:** 10.3390/genes13060980

**Published:** 2022-05-30

**Authors:** Quanshu Wu, Qiuyue Huang, Huilin Guan, Xiaoni Zhang, Manzhu Bao, Mohammed Bendahmane, Xiaopeng Fu

**Affiliations:** 1Key Laboratory of Horticultural Plant Biology, College of Horticulture and Forestry Sciences, Huazhong Agricultural University, Wuhan 430070, China; quanshuwu@webmail.hzau.edu.cn (Q.W.); huangqiuyue@webmail.hzau.edu.cn (Q.H.); guanhuilin@webmail.hzau.edu.cn (H.G.); zhangxiaoni@webmail.hzau.edu.cn (X.Z.); mzbao@mail.hzau.edu.cn (M.B.); 2Key Guangdong Laboratory for Lingnan Modern Agriculture, Genome Analysis Laboratory of the Ministry of Agriculture, Agricultural Genomics Institute at Shenzhen, Chinese Academy of Agricultural Sciences, Shenzhen 518120, China; 3Laboratoire Reproduction et Development des Plantes, UMR INRAE-CNRS-Lyon1-ENSL, Ecole Normale Supérieure, 69364 Lyon, France

**Keywords:** histone acetylation, gene expression, heat stress, *Rosa*

## Abstract

Roses have high economic values as garden plants and for cut-flower and cosmetics industries. The growth and development of rose plants is affected by exposure to high temperature. Histone acetylation plays an important role in plant development and responses to various stresses. It is a dynamic and reversible process mediated by histone deacetylases (HDAC) and histone acetyltransferases (HAT). However, information on HDAC and HAT genes of roses is scarce. Here, 23 HDAC genes and 10 HAT genes were identified in the *Rosa chinensis* ‘Old Blush’ genome. Their gene structures, conserved motifs, physicochemical properties, phylogeny, and synteny were assessed. Analyses of the expression of HDAC and HAT genes using available RNAseq data showed that these genes exhibit different expression patterns in different organs of the three analyzed rose cultivars. After heat stress, while the expression of most HDAC genes tend to be down-regulated, that of HAT genes was up-regulated when rose plants were grown at high-temperature conditions. These data suggest that rose likely respond to high-temperature exposure via modification in histone acetylation, and, thus, paves the way to more studies in order to elucidate in roses the molecular mechanisms underlying rose plants development and flowering.

## 1. Introduction

Histone acetylation is well known to be involved in different processes of plant growth and development [1,2,3], and in response to environmental changes [4], such as heat stress [5]. Histone acetylation has become an important part of epigenetics and has received considerable attention recently. Histone acetylation/deacetylation is a dynamic and reversible process mediated by HATs and HDACs. HDACs are involved in histones deacetylation, inhibiting transcription of functional genes, and, ultimately, silencing their expression [6,7]. In *Arabidopsis thaliana*, these HDACs can be divided into three subfamilies: RPD3/HDA1, HD2, and SIR2 [8]. Members of the HD2 subfamily are plant-specific HDACs [9]. HATs can neutralize the positive charge on lysine residues, weaken the binding effect of histone to DNA, induce relaxed chromatin structure, and facilitate the binding of transcription factors or transcriptional regulatory proteins to DNA, thus promoting gene transcription [10,11]. HATs can be divided into four subfamilies, named GNAT, MYST, CBP, and TAF_II_250 [8]. Although a number of studies, primarily using the plant model *Arabidopsis*, have shown that histone acetylation/deacetylation mediated by HAT and HDAC genes, is associated respectively with up- or down-regulation of genes, respectively, these processes are far from fully understood in plants in general.

In *Arabidopsis*, *HDA1* was strongly expressed in floral organs, and mutant plants *hda1* exhibit abnormal flower development [12,13]. HDA19, was reported to control *Arabidopsis* seed maturation and flowering time by inhibiting *AGL19*, and to promote flowering to some extent through the FLC pathway [14,15]. The floral organ identity gene *APETALA2* (*AP2*) was reported to regulate the expression of floral organ-related genes during flower development by recruiting co-repressors TOPLESS and HDA19 [16]. It was found that overexpression or knockdown of the RPD3-type HDAC gene *ZmHDA101* caused retarded plant growth and delayed flowering in transgenic maize [17]. Similarly, HAT enzymes were also shown to be involved in regulating flowering time and flower development. In *Arabidopsis*, CBP genes were shown to regulate the expression of *FLOWERING LOCUS C* (*FLC*), a major floral repressor, thereby promoting flowering [2]. Similarly, *HAG1/GCN5* gene plays essential roles in a number of plant developmental processes, such as leaf and floral organogenesis [18].

HDACs and HATs were also shown to play important roles in plant responses to exogenous stresses. *Arabidopsis* plants overexpressing HD2-type genes showed greater tolerance to abiotic stresses, such as drought, salt, and exposure to low or high temperature [19,20]. The transcription of *HD2C* was strongly induced at high temperature [20]. HAT genes were also shown to be involved in plant response to drought and salt stresses [21,22,23]. In maize, the expression of *ZmHATB* and *ZmGCN5* increases in plants exposed to 200 mM NaCl [22]. In rice, the expression of *OsHAC703*, *OsHAG703*, *OsHAM701*, and *OsHAF701* was also shown to be significantly up-regulated in drought stress [23].

As one of the 10 famous flowers in China, the rose is widely used in gardens, in the cut-flower industry and in cosmetics, making it of high economic value worldwide. The growth and development of rose plants are highly affected by high temperature, especially in the Yangtze River basin, China. Therefore, high temperatures seriously compromise ornamental and economic values of roses. Moreover, with the global climate changes, high temperatures occur frequently, making it difficult to plant roses. 

In this study, in order to lay the initial groundwork for the role of histone acetylation in response to heat stress in roses, we searched for genes coding for HDAC and HAT proteins using our high-quality assembled rose genome [24] and other available rose genome assemblies. We studied their structures, conserved motifs, physicochemical properties, phylogeny, and synteny. Additionally, the tissue-specific expression of the HDAC and HAT genes was analyzed in two *Rosa* species (*R. chinensis*, *R. wichurana*) and one modern rose cultivar (*R. hybrida* ‘Yesterday’). Finally, the expression levels of HDAC and HAT genes were evaluated in *R. chinensis* ‘Chilong Hanzhu’ plants grown under high-temperature conditions. The data provide insight into the molecular mechanisms underlying flower development in terms of epigenetics response to high temperature in *Rosa* spp. via expression of HDAC and HAT genes, and histone acetylation.

## 2. Materials and Methods

### 2.1. Heat Stress and Plant Materials Collection

*R. chinensis* ‘Chilong Hanzhu’ (CL) plants were obtained from Huazhong Agricultural University (Wuhan, China). For the heat-stress treatments, 30 individual plants were pruned and then randomly put into two incubators with two different parameters. ‘CL’ plants were heat-treated as follows: 16/8 h day/night at temperature 35 °C/30 °C, respectively. ‘CL’ plants control were grown at the following conditions: 16/8 h day/night at temperature 25 °C/20 °C, respectively. After 50 d, the leaves, sepals, petals, stamens, and pistils were harvested from heat-treated and control plants. Samples were harvested from five plants of each biological replicate. They were then combined, immediately frozen in liquid nitrogen, and then stored at −80 °C until analyses. 

### 2.2. Identification of the HDAC Genes and HAT Genes in Rose

Nucleotide sequence of *Arabidopsis* members of the HDAC and HAT genes were retrieved from *Arabidopsis* genome database (https://www.arabidopsis.org/, accessed on 1 June 2020) using TBtools [25] using available gene ID (Appendix A) [8]. *Arabidopsis* HDACs and HATs amino acid sequences were used to identify HDACs and HATs putative homologues in the genomes of *R. chinensis* [24], of *R. rugosa* [26], of *R. multiflora* [27], and of *Fragaria vesca* [28], using blast Wrapper with an E-value ≤ 1 × 10^−5^. Protein sequences of HDACs and HATs from *R. chinensis*, *F. vesca*, *R. rugosa*, *R. multiflora* were retrieved using NCBI blastp and Swiss-Prot database (BLAST: Basic Local Alignment Search Tool (nih.gov, accessed on 2 June 2020)). 

Conserved domains of HDAC and HAT proteins were detected by Batch CD-Search (https://www.ncbi.nlm.nih.gov/Structure/bwrpsb/bwrpsb.cgi, accessed on 2 June 2020) and then pictures were drawn with TBtools [25]. HDAC and HAT proteins with incomplete sequences were manually corrected. Candidate members were submitted to NCBI (https://www.ncbi.nlm.nih.gov/Structure/cdd/wrpsb.cgi, accessed on 5 June 2020) for the prediction of the structure of conserved domains.

### 2.3. Conserved Sequence and Structure Model Analysis

MEME (https://meme-suite.org/meme/tools/meme, accessed on 15 July 2021) was used to identify the conserved motifs in HDACs and HATs using the parameters of 20 motifs [29]. The structure information of the HDACs and HATs was confirmed using *R. chinensis* genome annotation data [24]. TBtools software v1.098696 (Guangzhou, China) [25] was used to visualize the distribution of HDACs and HATs.

### 2.4. Physicochemical Properties of HDACs and HATs

Expasy (https://web.expasy.org/protparam/, accessed on 18 July 2021) was used to predict the number of amino acid residues, theoretical isoelectric point, molecular weight, instability index classifiers, and grand average of hydropathicity (GRAVY) for identified rose HDAC and HAT proteins. 

### 2.5. Phylogenetic Analysis of HDACs and HATs

Phylogenetic trees were constructed with aligned HDAC and HAT protein sequences with Clustal W by using the Neighbour-Joining (NJ) method with 1000 iterations for the bootstrap values. The phylogenetic trees were generated using MEGA6. 

### 2.6. Chromosomal Location, Homologous Genes, and Synteny

The genome sequence and genes annotation information of *R. chinensis* [24], *R. Rugosa* [26] and *F. vesca* [28] were downloaded. TBtools software [25] was used to visualize the distribution of the HDAC and HAT genes on the *R. chinensis* chromosomes. Blastp and Quick Run MCScanX Wrapper were performed using TBtools [25] to analyze tandem repeats and synteny of the HDAC and HAT genes between *R. chinensis*, *R. rugosa*, and *F. vesca.*

### 2.7. Transcriptome Analysis of HDACs and HATs

Published RNA-seq data of *R. chinensis* ‘Old Blush’ (OB) [30,31], *R. wichurana* (Rw) and one modern rose cultivar *R. hybrida* ‘Yesterday’ (Ry) (accession number: PRJNA436590) were downloaded from the NCBI (https://www.ncbi.nlm.nih.gov/, accessed on 5 February 2021) to monitor the expression patterns of the HDACs and HATs. Heat maps were drawn by TBtools [25]. 

### 2.8. RNA Preparation and qRT-PCR Analysis 

Total RNA was extracted using an EASYspin Plant RNA extraction kit (Aidlab, Beijing, China). TRUEscript RT MasterMix (Aidlab, Beijing, China) was used for reverse transcription using 1 μg of RNA. Primer 5 was used to design primers for quantitative real-time PCR (qRT-PCR) (Appendix A). The qRT-PCR was performed using PowerUp SYBR Green Master Mix (Applied Biosystems, Carlsbad, CA, USA) in 384-well plates. An Applied Biosystems Real-Time PCR System (Life Technologies, Carlsbad, CA, USA) was used with cycling parameters: heating at 95 °C for 2 min, 40 cycles of denaturation at 95 °C for 10 s, annealing at 60 °C for 20 s, and extension at 72 °C for 35 s. Three biological replicates were performed for each sample in order to confirm the reliability of the results. The *RcGAPDH* and *RcACTIN* genes [32,33] were used as an internal quantitative control to normalize samples (Appendix A). The relative expression values were calculated using the comparative CT(2^−ΔΔCT^) method [34]. Statistical analysis was performed using single factor ANOVA along with Least Significant Difference test using SAS 9.4, and *p* < 0.05 were considered significant.

## 3. Results

### 3.1. The Identification of HDACs and HATs

HDAC and HAT gene members were retrieved in the six species (*R. chinensis*, *R. rugosa*, *R. multiflora*, *F. vesca*, *A. thalian**a*, and *Oryza sativa;*
Table 1) based on previous reports or by BLAST search. 

In *F. vesca*, alternative splicing transcript variants were identified for a number of HDAC and HAT genes, all of which retained the t1 suffix for further analyses. The number of HDAC and HAT genes varied among the six species (Table 1). Blastp and conserved domain search identified 23 HDACs in *R. chinensis*, 26 HDACs in *R. multiflora*, 12 HDACs in *R. rugosa* and 14 HDACs in *F. vesca*. Concerning the HAT genes, 10 were identified in *R. chinensis*, 15 in *R. multiflora*, 11 in *R. rugosa*, and 13 in *F. vesca* (Appendix A). They were named according to the naming conventions in *Arabidopsis* [8].

Compared with *Arabidopsis*, *R. chinensis* has a higher number of HDAC genes, and lower number of HAT genes (Table 1). In addition, the number of RPD3/HDA1 subgroup members varied greatly between the rose and *F. vesca*, with 19 members in *R. chinensis* and only 10 members in *F. vesca*. Only 10 RPD3/HDA1 subgroup members were found in *R. rugosa.* Similarly, while six members of HD2 subfamily were identified in *R. multiflora*, only two members were identified in *R. chinensis* and only one member was identified in *R. rugosa.* The differences in HDAC and HAT gene numbers (Table 1) could be due to the possibility that these rose species may have evolved independently regarding the process of histone acetylation. This is supported for example by the recent report that proposed that *R. rugosa* diverged from *R. chinensis* and over 6 million years ago [35].

**Table 1 genes-13-00980-t001:** Number of HDAC and HAT genes in different species.

Family	Subfamily	*A. thaliana*	*O. sativa*	*R. multiflora*	*R. rugosa*	*R. chinensis*	*F. vesca*
HDAC	RPD3/HDA1	12	14	18	10	19	10
	SIR2	2	2	2	1	2	2
	HD2	4	3	6	1	2	2
HAT	GANT	3	3	6	5	4	7
	MYST	2	1	3	1	1	1
	CBP	5	3	3	3	3	3
	TAF_II_250	2	1	3	2	2	2
Reference		[8]	[36,37]	This study	This study	This study	This study

### 3.2. Gene Structure and Conserved Domains

Structure analyses of coding sequences showed that HDAC genes have different numbers of exons and introns. In *R. chinensis*, the SIR2 subfamily genes have high number of introns, while the HD2 subfamily genes have fewer introns (Figure 1). The number of introns in genes of *R. chinensis* also differed from that of *A. thaliana* and *F. vesca* (Figure 1 and Appendix A). Genes of the RPD3/HDA1 subfamily had the highest number of introns, reaching with seventeen introns in the case of *RcHDA15*, although some members have no or only one or two introns. HD2 subfamily genes of *A. thaliana* and *F. vesca* have higher number of introns compared to *R. chinensis* (Figure 1 and Appendix A), while the number of introns in RPD3/HDA1 and SIR2 subfamilies were similar in *R. chinensis*, in *A. thaliana*, and in *F. vesca*, with the highest number of introns reaching 17 in *RcHDA15*, *AtHDA15*, and *FvHDA15*. To some extent, this indicated that the RPD3/HDA1 subfamily and SIR2 subfamily were conserved among the studied species here, while the number of introns in the HD2 subfamily of rose was different from that of *Arabidopsis* and strawberry. The number of exons and introns also varied between HAT genes of *R. chinensis*, of *A. thaliana* and of *F. vesca*. (Appendix A). For example, *RcHAC4*, *RcHAF2*, *FvHAC4*, and *FvHAF2* had fewer introns than *AtHAC4* and *AtHAF2*.

Motifs distribution of the 23 RcHDAC and the 10 RcHAT proteins were investigated using the MEME 5.3.3 program. In total, 20 conserved motifs were identified (Figure 1 and Appendix A). The number of motifs varied among RPD3/HDA1 subfamily members. Motif e was unique in RcHDA6-1, RcHDA6-2, RcHDA9, and RcHDA19. Motif a was found in all the RPD3/HDA1 subfamilies except RcHDA2-2, RcHDA2-3, RcHDA2-4, RcHDA2-5, RcHDA2-6, RcHDA2-7, and RcHDA2-8. Although, RcHAG1, RcHAG2, and RcHAG4 all have complete conserved domains, no motifs could be found in these proteins (Appendix A). Motifs 1 to 9 were found in all CBP members, and motifs 11 to 20 were found in all members of TAF_II_250.

The conserved domains in the 23 RcHDACs and in the 10 RcHATs were analyzed by NCBI-CDD (Appendix A). All members of the RPD3/HDA1 subfamily possess the characteristic domains of HDAC family. The two members of the SIR2 subfamily, RcSRT1 and RcSRT2, contain the conserved domains SIRT4 or SIRT7, respectively. HD2 members have a highly conserved NPL domain, whereas the GNAT subfamily proteins RcHAG1, RcHAG2, and RcHAG3 (includes RcHAG3-1 and RcHAG3-2) have 3 different conserved domains, COG5076, Hat1_N and ELP3, respectively. RcHAG4 have the conserved domain of MYST, designated PLN00104. The members of CBP all have a conserved HAT_KAT11 domain, and TAF_II_250 members have a highly conserved DUF3591 domain. 

### 3.3. Physicochemical Properties of HDACs and HATs

Expasy was used to analyze the physicochemical properties of RcHDACs and RcHATs (Table 2). The size of *R. chinensis* HDAC proteins varied greatly, with the largest protein composed of 661 amino acids residues and the smallest composed of 46 amino acids residuces. The theoretical PI of SIR2 subfamily is about 9. The theoretical PI of RPD3/HDA1 and HD2 subfamily members were discrepant, and there were great differences among subfamily members. In terms of protein stability, SIR2 subfamily proteins are predicted to be unstable proteins, and HD2 subfamily proteins are predicted as stable proteins. In total, 11 members of the RPD3/HDA1 subfamily are predicted to be stable proteins (i.e., RcHDA14-2), and 8 are predicted to be unstable proteins (ie. RcHDA2-6). SIR2 subfamily proteins were all hydrophilic, while members of the HD2 and RPD3/HDA1 subfamilies were more hydrophobic. Hydrophilic proteins such as RcHDA6-2 and RcHDA19 were also present in the RPD3/HDA1 subfamily.

RcHATs proteins were all hydrophilic. The size of RcHAT proteins varied greatly, with the largest protein containing 1900 amino acid residues and the smallest protein containing 95 amino acid residues. All HAT proteins, except RcHAG3-1 and RcHAG3-2, were rated as likely unstable proteins. 

### 3.4. Phylogenetic Relationships of HDACs and HATs

#### 3.4.1. RPD3/HDA1 Subfamily

To know more about the phylogenetic relationships among RPD3/HDA1 genes, a phylogenetic analysis of RPD3/HDA1 proteins from *R. chinensis* (Appendix A), from *A. thaliana* (Appendix A) and from *F. vesca* (Appendix A) was constructed by NJ method. RcRPD3/HDA1 proteins and FvRPD3/HDA1 proteins were divided into 4 clades named Class I to Class IV with reference to the classification of the *A. thaliana* (Figure 2a). While *R. chinensis*, *F. vesca* and *A. thaliana* have comparable numbers of gene member for Class I and Class II, *R. chinensis* has more gene members for Class III (8 genes for *R. chinensis* to one in *F. vesca* and one in *A. thaliana*) and Class IV (5 genes for *R. chinensis*, 2 in *F. vesca* and 2 in *A. thaliana*).

#### 3.4.2. HD2 Subfamily

Phylogenetic analysis of *R. chinensis* HD2 subfamily proteins was constructed with *F. vesca* and *A. thaliana* HD2 proteins by NJ method (Figure 2b). RcHDT1 and RcHDT2 are more closely related to FvHDT3-1 and FvHDT3-2, respectively, than to the HD2 proteins of *A. thaliana*. The number of HD2 subfamily proteins varied among species, with 4 members in *A. thaliana*, and 2 members in *R. chinensis* and *F. vesca*.

#### 3.4.3. SIR2 Subfamily

*R. chinensis*, *F. vesca*, and *A. thaliana* have two members each of SIR2 protein. Phylogenetic analysis shows that the SIR2 subfamily is highly conserved among the 3 species (Figure 2c).

#### 3.4.4. HAT Family

Phylogenetic analysis divided HAT proteins into four subfamilies: MYST, CBP, TAF_II_250, and GNAT (Figure 2d). The GNAT subfamily was previously divided into four classes designated as GCN5, ELP3, HAT1, and HPA2 [8]. In *Arabidopsis*, the GCN5, ELP3, and HAT1 classes are each composed of a single gene (AtHAG1, AtHAG3, and AtHAG2, respectively), whereas no members of the HPA2 class have been identified in *A. thaliana* [8]. Here, we found that *R. chinensis* and *F. vesca* HAT proteins can be divided into 6 clades, namely GCN5, HAT1, ELP3, MYST, CBP, and TAF_II_250. A GCN5 homolog and a HAT1 homolog were identified in *R. chinensis* (RcHAG1 and RcHAG2, respectively) and in *F. vesca* (FvHAG1 and FvHAG2, respectively), thus similar to *Arabidopsis*. Two and 5 ELP3 homologs were identified in the rose (RcHAG3-1, RcHAG3-2) and in *F. vesca* (FvHAG3-1, FvHAG3-2, FvHAG3-3, FvHAG3-4, and FvHAG3-5), respectively. *R. chinensis* also has one member of the MYST clade, two members of the TAF_II_ 250 clade and 3 members of the CBP clades.

### 3.5. Chromosomal Location, Homologous Genes, and Synteny

The 23 HDAC and 10 HAT genes are distributed on the 7 chromosomes of the rose (Figure 3a).

Synteny analysis with *R. rugosa* and *F. vesca*, two closely related species to *R. chinensis*, revealed that homologs of 10 HDAC genes were found in *F. vesca* genome and nine in *R. rugosa* genome (Figure 3b; indicated with orange lines). Nine HAT homologous genes were found in *F. vesca* and in *R. rugosa* genomes (Figure 3b, blue lines). The remaining members of HDACs, such as *RcHDA14-3*, *RcHDA2-2*, and *RcHDA2-3* found on chromosomes 1 and 4, could not be found in *R. rugosa* and *F. vesca* genomes (Figure 3b). It is possible that these HDACs are unique to *Rosa* compared to *Fragaria*. The absence of these HDACs in the *R. rugosa* genome is likely related to the fact that the available genome sequence of *R. rugosa* is not be complete to identify these genes. 

### 3.6. Expression of HDACs and HATs in Different Organs of Three Roses

The expression patterns of HDACs and HATs in various organs of 2 *Rosa* species (*R. chinensis*, *R. wichurana*) and 1 modern rose cultivar (*R. hybrida* ‘Yesterday’) were analyzed. Expression analysis using published RNAseq data [30,31] shows that *RcHDA6-1*, *RcHDA2-6*, *RcHDA2-7*, *RcHDA2-8*, *RcHDA14-2*, *RcHDA14-3*, and *RcHDA14-4* are not expressed in all tested tissues of the three roses (Figure 4). HDAC genes have different expression levels depending on the organ studied. For example, *RcHDA6-2*, *RcHDA14-1*, and *RcSRT**2* are highly expressed in roots, leaves, and pink petals of *R. chinensis* ‘OB’, respectively (Figure 4a). Whereas high transcript abundance for *RcHDA19*, *RcHDT1*, and *RcHDT2* were observed in green petals of *R. chinensis* ‘OB’. In ‘Rw’, *RcHDA19*, *RcHDT1*, and *RcHDT2* were highly expressed in flower buds, a stage where petals are still green, thus similar to ‘OB’. *RcHDA6-2* showed the highest transcript abundance in stems of ‘Rw’ and ‘Ry’ roses, while *RcHDA14-1* was highly expressed in leaves (Figure 4b). These data shows HDAC genes expressed are expressed in different tissues and organs of the rose, and suggest their potential roles in vegetative and reproductive growth.

According the RNAseq data [30,31] HAT genes also show different expression levels in different tissues of the rose. The 2 HAT genes *RcHAG3-2* and *RcHAC4* are likely not expressed in roots and stems of Rw’ and ‘Ry’ roses (Figure 5), while *RcHAG3-1* is highly expressed in developing petals at onst of color change in ‘OB’ flower buds (Figure 5a). *RcHAG2* showed the highest transcript abundance in flower buds of ‘Rw’ rose, respectively (Figure 5b, Appendix A), and transcript of *RcHAC2* highly accumulated in roots of ‘Rw’ and ‘Ry’ roses. The fact that HAT genes are expressed in different tissues of the rose suggest their likely roles in histone acetylation during plant organs development.

### 3.7. Responses of RcHDACs and RcHATs to High Temperature

*R. chinensis* ‘Chilong Hanzhu’ (‘CL’) plants were exposed to high-temperature stress (35 °C/30 °C, day/night). Sepals, petals, stamens, pistils, and leaves sample were then collected from heat-treated and control plants for qRT-PCR experiments (Figure 6).

qRT-PCR analyses showed that in ‘CL’ roses, the class I gene *RcHDA6-1* was highly expressed in leaves and pistils of the control plants, and following high-temperature treatments, its expression was significantly decreased in these organs, while its expression was increased in stamens (Figure 6; Appendix A). The expression of *RcHDA15* (Class II) and *RcHDA2-2*(Class III) also decreased in leaves and sepals in response to high-temperature stress. *RcHDA2-1*(Class III), *RcHDA2-5*(Class III), *RcHDA8*(Class IV), and *RcHDA14-2* (Class IV) also decreased after high-temperature treatments in leaves, pistils, and sepals. *RcSTR2* (SIR2 subfamily) was expressed in the leaves, and its expression level also decreased after high-temperature treatment. HD2 subfamily members, *RcHDT1* and *RcHDT2*, were expressed in almost all tissues analyzed, with the highest expressed level in the leaves or stamen, respectively (Appendix A). Their expression also decreased after high-temperature treatment. 

In contrast to HDAC genes, the expression of HAT genes *RcHAG1*, *RcHAG2*, and *RcHAF2* increased after high-temperature treatments in stamens, pistils, and leaves (Figure 6b, Appendix A). Similarly, the expression of *RcHAC4* was also up-regulated in petals, stamens, and leaves after high-temperature treatments (Figure 6; Appendix A). *RcHAG4* was expressed in a variety of tissues, and after high-temperature treatments, its expression was up-regulated in sepals, stamens, pistils, and leaves.

These data show that, overall, the expression levels of most genes of the HDACs were down-regulated after high-temperature treatments, especially in leaves, whereas that of most HAT genes was up-regulated after high-temperature treatment, especially in leaves.

## 4. Discussion

### 4.1. The Expression Specificity of HDACs and HATs 

With the decoding of more and more plants genomes, HDAC and HAT genes families were identified in many plants, such as in *Arabidopsis* [8], in rice [36,37], in tomato [39], etc. 

Histone acetylation plays an important role in plant development and stress responses. These are dynamic and reversible processes mediated by HDACs and HATs enzymes. Our data show that few HDAC and HAT genes with FPKM values greater than 15 exhibit expression patterns in different tissues and organs (i.e., stems, leaves, and flower buds), and in different rose cultivars (Figure 7). We focused on genes for which FPKM values were greater than 15 in stamens, in carpels, in green petals (flower bud stage), in petals changing colors, or in fully developed pink petals of ‘OB’. 

The HDAC genes, *RcHDA6-2*, *RcHDA19*, and *RcHDT1* were strongly expressed in roots, stems of ‘OB’, ‘Rw’, and ‘Ry’, while *RcHDA14-1* was only strongly expressed in the leaves. All members, except *RcHDA6-1*, of class I of RPD3/HDA1 subfamily were expressed in the three rose cultivars. Previous report showed that *HDA6* acts as a repressor for EIN3-mediated transcription and Jasmonate signaling and suggested that *HDA6* was involved in root development [40,41]. *RcHDA6-2* was strongly expressed in roots of ‘OB’, suggesting a putative role in root development in roses, thus in agreement with previously reported data in *Arabidopsis*. 

Interestingly, many RPD/HDA1 members, including *RcHDA6-1*, *RcHDA2-6*, *RcHDA2-7*, *RcHDA2-8*, *RcHDA14-2*, *RcHDA14-3*, and *RcHDA14-4*, showed very low to no expression in all analyzed organs. Either, these genes are not expressed in these organs, or are not expressed at the analyzed developmental stage and/or growth conditions. *RcHDA14-1* was the only class IV gene that showed expression in the leaves of the three rose cultivars ‘OB’, ‘Rw’, and ‘Ry’, although its exact role in the leaves remains to be elucidated, requiring further studies.

Among the two SIR2 analyzed genes, only *RcSRT2* was expressed in ‘OB’ rose petals during flower opening (Figure 4). Recently, Zhang et al. showed that histone deacetylases SRT1 and SRT2 mediate ethylene-induced transcriptional repression by interacting with the protein EIN2 NUCLEAR ASSOCIATED PROTEIN1 (ENAP1) in seedlings of *Arabidopsis* [42]. It was reported in roses that ethylene influence flower opening by affecting ethylene receptor and/or signaling genes [42,43]. Therefore, it is possible that in the rose, *RcSRT1* and *RcSRT2* play a role in flower opening by acting in the ethylene pathway. Therefore, it will be interesting to address the link between *RcSRT1* and *RcSRT2* and ethylene perception and signaling during flower opening.

Members of the HD2 subfamily were expressed in roots, stems, leaves and flowers, with the highest expression level in flower buds and flowers (Figure 4), especially in young petals in closed buds, in petals at the changing colors development stage and in carpels (FB_GP, FB_CP, and Ca, respectively; Figure 4a). These data indicate that HD2 may play important roles in a diversity of functions during flower development. As support to this hypothesis, in *Arabidopsis*, *HDT1*, *HDT2*, and *HDT3* were shown to be important for ovule and embryo development [44], in longan, enhanced expression of *DlHD2* was associated with delayed fruit senescence [45], and in *O. sativa*, overexpression of *OsHDT1* was shown to promote early flowering by affecting key flowering time genes [46]. 

In the GNAT subfamily, *RcHAG1*, *RcHAG2*, *RcHAG3-1*, and MYST subfamily gene *RcHAG4* showed pleiotropic expression in most of the analyzed organs, while *RcHAG3-2* showed low to no expression (Appendix A). In *A. thaliana*, *HAG1* was shown to play an important role in flower development. *Arabidopsis* lines mutant in *HAG1* displayed a variety of pleiotropic growth defects including dwarfism, loss of apical dominance and loss of fertility [47], as well as homeotic conversion of petals into stamens and of sepals into filamentous structures and formation of ectopic carpels [48].

The expression of two CBP genes *RcHAC1* and *RcHAC4* was very low in the 3 rose cultivars. In *Arabidopsis*, *HAC1* was reported to play a regulatory role in flowering time and in development. Plants knockout for *AtHAC1* exhibit delayed flowering and partially reduced fertility [49], and *AtHAC1* was shown to function synergistically with *AtHAC4*, *AtHAC5*, and *AtHAC12* [50]. The expression of RcHAC genes also varied depending on the rose cultivar, thus, indicating putative functional differentiation in different cultivars.

### 4.2. The Antagonistic Expression Patterns of HDACs and HATs

HDACs and HATs play an important role in plant response to stresses, such as high temperature and drought. Roses have a great economic importance worldwide. Their economic value is mainly related to the characteristics of the flower. Here, our data show that HDACs and HATs have different expression patterns in the flower and in the leaf of the three analyzed rose cultivars.

High-temperature treatments of the ancient Chinese rose ‘CL’ showed that there were differences in gene expression trends between treated and untreated plants. The expression of most HDAC genes was down-regulated after high-temperature treatment in both floral organs and leaves, with a strong down-regulation observed in leaves. In contrast to *RcHDA6-2*, *RcHDA6-1* expression was high in pistils and leaves of ‘CL’ plants, and upon high-temperature treatments its expression was significantly decreased. This difference in expression patterns and response to high temperature of *RcHDA6-1* and *RcHDA6-2* may be related to the fact that *RcHDA6-2* is expressed at very low level in ‘CL’ plants. In *Arabidopsis*, *AtHDA6* has been shown to be involved in the response to salt stress [51]. *Arabidopsis* lines mutant for *AtHDA6* were hypersensitive to salt stress, indicating that expression of this gene is important for the salt stress response. An opposite tend appears to occur in the rose, where heat stress correlates with the down-regulation of *RcHDA6-1*. It is possible that this gene has evolved differently for the response to heat stress in different species. A similar behavior was observed for *RcHDA2-1* and *RcHDA14-2* whose expression was high in leaves and respectively in pistils or in sepals, and their expression was down-regulated upon heat treatments.

Conversely to other HDACs subfamilies, HD2 genes *RcHDT1* and *RcHDT2* were expressed in all analyzed organs, thus similar to previous report in *A. thaliana*, where *AtHD2C* was shown to be constitutively expressed in all organs [52]. Similar to other HDACs, the expression of both HD2 genes was down-regulated after heat treatment, indicating that both genes may be involved in the response to heat stress. In *A. thaliana*, HD2 subfamily genes have also been shown to be involved in plant development [19,53] and their expression was repressed by abscisic acid and was associated with decreased sensitivity to salt stress and drought [52]. Our data combined with those of the literature show the importance of HD2 genes in the response to various stresses such as heat, drought, salinity, and cold in plants. 

Conversely to HDAC genes, the expression of HAT genes was up-regulated after heat treatments. The highest response was observed in leaves, thus similar to HDACs, but in an opposite way. Previous studies have shown that HATs are involved in plant development, especially in flower development [49,50,52]. Interestingly, after high-temperature treatment, the expression of *RcHAG1*, *RcHAG2*, and *RcHAF2* was significantly up-regulated in stamens and pistils. Previous study have shown in roses that a high-temperature treatment affect meiosis, and, more specifically, gamete ploidy, leading to diplogametes formation [54]. Taken together with our data, it possible that HAT genes *RcHAG2*, *RcHAG2*, *RcHAG4*, and *RcHAF2* have a role in gametes formation, although the direct link between their miss expression in response to high temperature and the formation of diplogamete in roses has not yet been established.

HDACs and HATs jointly regulated histone acetylation, and the expression trends of HDACs and HATs showed opposite changes after high-temperature treatment (Figure 8), indicating that the histone acetylation degrees in rose flowers and leaves are enhanced under high-temperature treatment, which may be one of the ways to cope with high-temperature stress.

Moreover, since histone acetylation plays an important role in plant growth and development and resistance to various stress, the fact that the expression levels of histone acetylation-related genes families are modified in rose grown in high-temperature environments, may indicate their important roles in response to heat stress. Therefore, the present work paves the way for further studies to help identify the associated regulatory pathways and specific molecular mechanisms. 

## 5. Conclusions

In this study, 23 HDAC and 10 HAT genes were identified in *R. chinensis* ‘Old Blush’. Their structures, conserved motifs, physicochemical properties, phylogeny and synteny were studied. The tissue-specific expression of the HDAC and HAT genes was analyzed in three rose cultivars (*R. chinensis* ‘Old Blush’, *R. wichurana* and *R. hybrida* ‘Yesterday’). After heat stress, it showed the antagonistic expression patterns of HDACs and HATs in *R. chinensis* ’Chilong Hanzhu’. This study paved a way for deciphering the functional roles of RcHDAC and RcHAT genes during plant growth as well as addressing the link between histone acetylation and heat stress response in roses.

## Figures and Tables

**Figure 1 genes-13-00980-f001:**
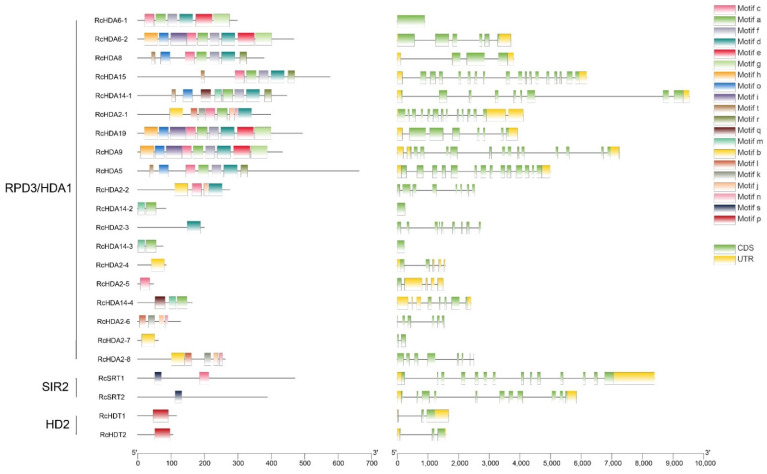
Conserved protein motifs (**left**) and gene structures (**right**) analyses of the 23 *R. chinensis* HDACs. (**left**) panel shows protein motifs a to t indicated by different color boxes. (**right**) panel: Gene structure showing UTR indicated by yellow and CDS with introns indicated by yellow and green boxes, respectively. Solid lines indicate introns.

**Figure 2 genes-13-00980-f002:**
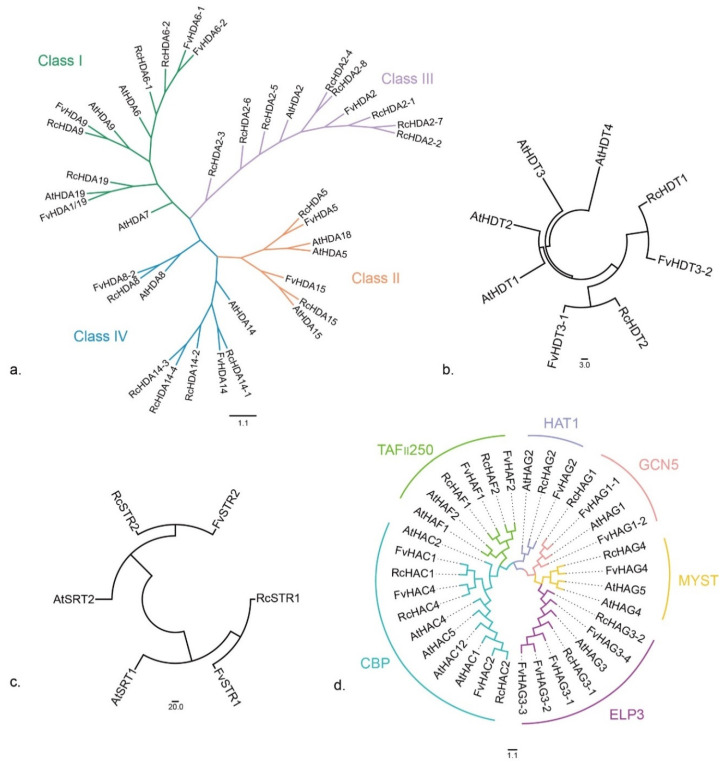
Phylogenetic analysis of HDAC proteins and HAT proteins constructed by the neighbor-joining (NJ) method in MEGA6. (**a**) Phylogenetic tree of *R. chinensis*, *F. vesca and A. thaliana* RPD3/HDA1. The 4 clades were named Class I to Class IV, and they are marked in different colors. (**b**,**c**) Phylogenetic tree of *R. chinensis*, *F. vesca* and *A. thaliana* STR2 (**b**), HD2 (**c**) and HAT (**d**) proteins. The 6 clades are marked in different colors.

**Figure 3 genes-13-00980-f003:**
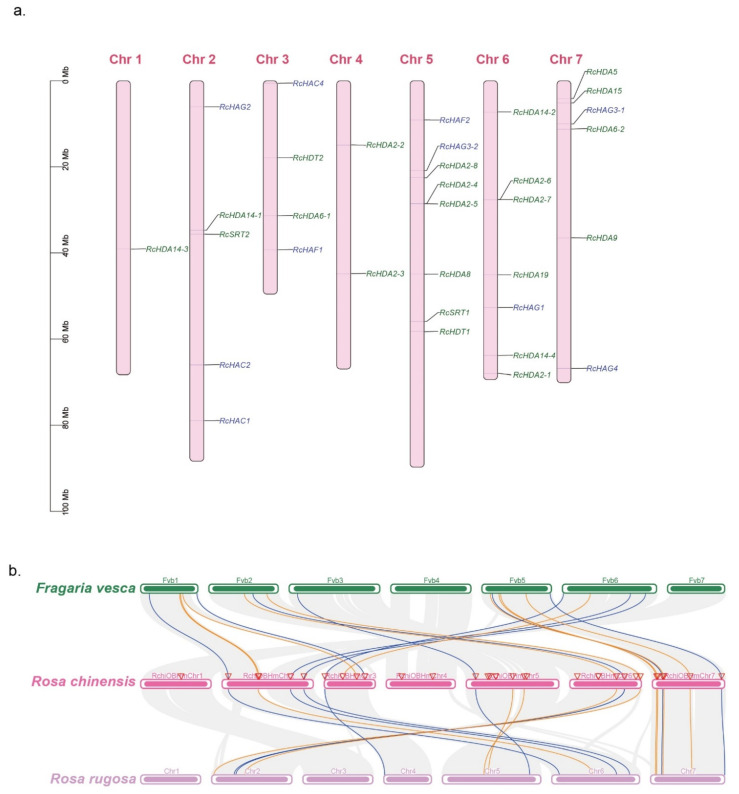
Genome distribution of HDAC and HAT genes in *F. vesca*, *R. chinensis* and *R. rugosa*. (**a**) *RcHDACs* and *RcHATs* distribution map on the chromosomes of *R. chinensis*. HDACs are written in green color, and HATs are indicated in blue color. (**b**) HDAC and HAT genes synteny analysis between *R. chinensis*, *F. vesca* and *R. rugosa*. Orange and blue lines indicate synteny of HDACs and HATs, respectively.

**Figure 4 genes-13-00980-f004:**
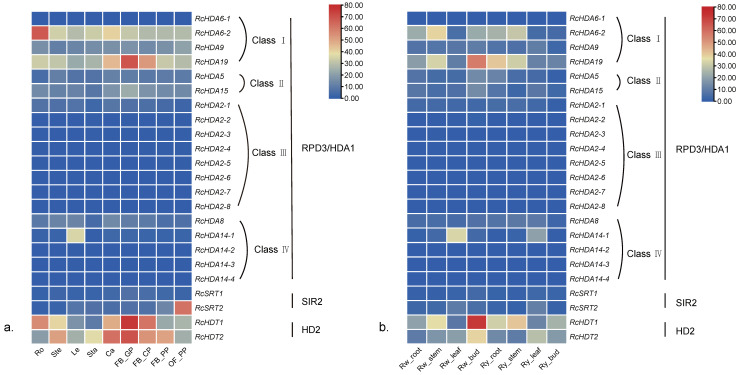
Expression patterns of HDAC genes in 2 *Rosa* species (*R. chinensis*, *R. wichurana*) and 1 modern rose cultivar (*R. hybrida* ‘Yesterday’). (**a**) Heatmap expression of 23 HDAC genes in different organs of *R. chinensis* ‘Old Blush’ (OB). Ro: roots; Ste: stems; Le: leaves; Sta: stamens; Ca: carpels; FB_GP: green petals in the flower buds; FB_CP: petals changing colors in the flower buds; FB_PP: pink petals in the flower buds; and OF_PP: pink petals on the open flowers. (**b**) Organ specific expression of the 23 HDAC genes in *R. wichurana* (Rw) and in the modern rose cultivar *R. hybrida* ‘Yesterday’ (Ry).

**Figure 5 genes-13-00980-f005:**
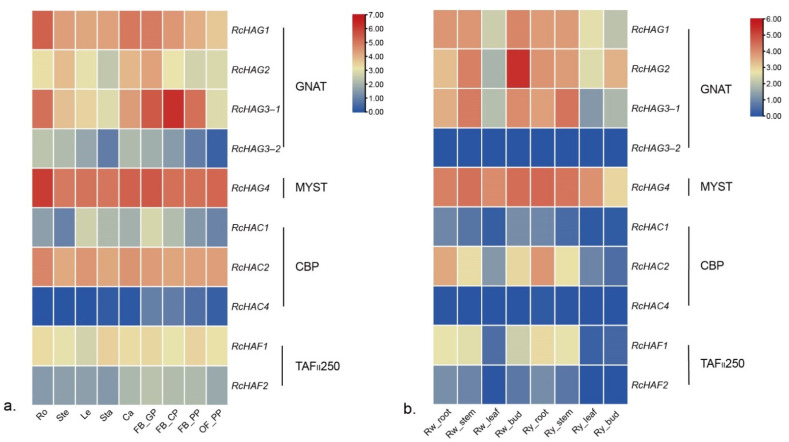
Expression patterns of HAT genes in *R. chinensis*, *R. wichurana*, and *R. hybrida* ‘Yesterday’. Up- and down-regulated transcripts are indicated, respectively, in red and blue (log2-fold change). (**a**) Heatmap of expression of 10 HAT genes in *R. chinensis* ‘Old Blush’(OB). Ro: roots; Ste: stems; Le: leaves; Sta: stamens; Ca: carpels; FB_GP: green petals in the flower buds; FB_CP: petals changing colors in the flower buds; FB_PP: pink petals in the flower buds; and OF_PP: pink petals on the open flowers. (**b**) Heatmap of expression of 10 HAT genes in *R. wichurana* and *R. hybrida* ‘Yesterday’.

**Figure 6 genes-13-00980-f006:**
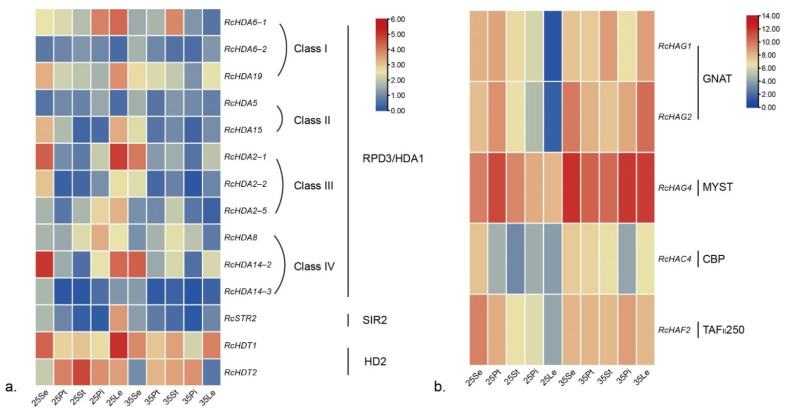
Heatmap of expression of selected HDAC (**a**) and HAT (**b**) genes in *R. chinensis* ‘Chilong Hanzhu’ (CL) roses grown under control condition (25 °C) or at high temperature (35 °C) condition (log2-fold change). 25Se, 25Pt, 25St, 25Pi, and 25Le represent, respectively, sepal, petal, stamen, pistil, and leaf samples harvested from plants grown under 25 °C/20 °C (day/night). 35Se, 35Pt, 35St, 35Pi, and 35Le correspond to similar samples but harvested from plants grown at 35 °C/30 °C (day/night).

**Figure 7 genes-13-00980-f007:**
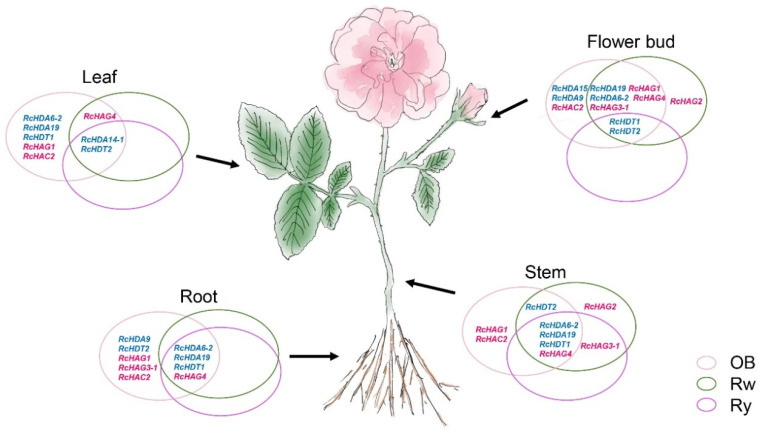
Summary of expression patterns of the HDAC (in blue) and HAT (in red) genes in various organs of the three rose cultivars, *R. chinensis* ‘Old Blush’ (OB), *R. wichurana* (Rw), and *R. hybrida* ‘Yesterday’ (Ry). The Venn diagrams show the expression of HDAC and HAT genes in the 3 rose cultivars: pink ellipse = in ‘OB, green ellipse = in ‘Rw’, and purple ellipse = in ‘Ry’.

**Figure 8 genes-13-00980-f008:**
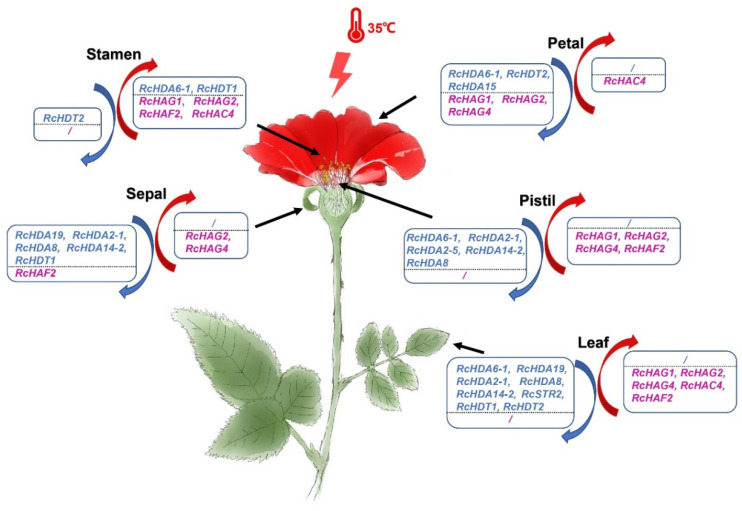
Expression patterns of the HDAC (blue) and HAT (red) genes in various organs of *R. chinensis* ‘CL’ grown in normal or at high-temperature conditions. Boxes to the left with blue downward-pointing arrow contain genes whose expression level was down-regulated in plants growth at high temperature. Boxes to the right with red upward-pointing arrow contain genes whose expression levels were up-regulated in plants growth at high-temperature treatment.

**Table 2 genes-13-00980-t002:** Physicochemical properties of *R. chinensis* HDACs.

Subfamily	Name	Size(aa) ^1^	PI	MW	Ii	Classifies	GRAVY
RPD3/HDA1	RcHDA6-1	297	6.05	33,956.99	44.54	unstable	−0.191
	RcHDA6-2	466	5.33	52,454.87	45.42	unstable	−0.514
	RcHDA8	377	5.63	41,217.82	29.37	stable	−0.166
	RcHDA15	575	5.97	62,865.81	37.96	stable	−0.328
	RcHDA14-1	445	5.96	48,359.75	37.04	stable	−0.137
	RcHDA2-1	397	9.04	44,059.8	43.81	unstable	−0.038
	RcHDA19	492	5.14	55,455.22	41.09	unstable	−0.531
	RcHDA9	432	5.38	49,400.64	37.07	stable	−0.404
	RcHDA5	661	5.29	73,513.29	40.76	unstable	−0.291
	RcHDA2-2	274	8.93	31,459.6	44.02	unstable	−0.131
	RcHDA14-2	83	5.83	9359.63	16.39	stable	0.294
	RcHDA2-3	198	7.61	22,437.29	38.39	stable	0.432
	RcHDA14-3	75	5.12	8463.73	14.47	stable	0.611
	RcHDA2-4	84	5.40	9646.11	27.87	stable	−0.029
	RcHDA2-5	46	7.66	5062.87	34.16	stable	0.004
	RcHDA14-4	162	9.11	18,202.9	37.42	stable	−0.154
	RcHDA2-6	127	9.39	14,334.89	51.03	unstable	0.130
	RcHDA2-7	61	5.81	7012.15	24.68	stable	−0.051
	RcHDA2-8	261	6.54	28,953.16	40.33	unstable	−0.182
SIR2	RcSRT1	469	9.10	52,092.43	46.23	unstable	−0.191
	RcSRT2	387	9.54	42,858.86	44.43	unstable	−0.290
HD2	RcHDT1	115	4.70	12,675.34	29.21	stable	0.050
	RcHDT2	105	9.12	11,575.43	29.42	stable	0.000
GNAT	RcHAG1	547	6.27	61,085.47	46.09	unstable	−0.597
	RcHAG2	460	5.12	51,748.95	41.74	unstable	−0.224
	RcHAG3-1	567	8.74	63,437.87	30.76	stable	−0.333
	RcHAG3-2	95	10.12	10,489.60	30.19	stable	−0.071
MYST	RcHAG4	470	6.66	54,232.93	41.66	unstable	−0.583
CBP	RcHAC1	1377	6.48	154,650.31	57.02	unstable	−0.477
	RcHAC2	1717	8.46	194,137.68	57.04	unstable	−0.753
	RcHAC4	955	7.22	107,704.14	44.62	unstable	−0.413
TAF_II_250	RcHAF1	1900	5.74	214,349.67	51.34	unstable	−0.822
	RcHAF2	1692	5.54	191,691.06	52.75	unstable	−0.724

Note: ^1^: Number of amino acid residues; PI: Theoretical isoelectric point; MW: Molecular weight; Ii: Instability index; and GRAVY: Grand average of hydropathicity. The GRAVY value for a peptide or protein is calculated as the sum of hydropathy values of all the amino acids, divided by the number of residues in the sequence [38].

## Data Availability

Genomic data of *R. chinensis* ‘Old Blush’ can be downloaded from https://lipm-browsers.toulouse.inra.fr//pub/RchiOBHm-V2/, accessed on 25 May 2020 [24]. Genomic data of *R. rugosa* can be downloaded from http://eplant.njau.edu.cn. Genomic datas of *Rosa multiflora and Fragaria vesca* can be downloaded from https://www.rosaceae.org/, accessed on 27 May 2020 [55]. Published RNA-seq data of *R. chinensis* ‘Old Blush’(OB) [30,31], *R. wichurana* (Rw),ratepe and *R. hybrida* ‘Yesterday’ (Ry) (accession number: PRJNA436590) can be downloaded from the NCBI (https://www.ncbi.nlm.nih.gov/, accessed on 5 February 2021).

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
