# Peer review of "Comprehensive Genome-Wide Analysis of Histone Acetylation Genes in Roses and Expression Analyses in Response to Heat Stress"

_genes, 2022, doi:10.3390/genes13060980_

Round 1
Reviewer 1 Report
In this study, HDACs and HATs from Rose genome were identified and many biochemical analyses were performed for the first time. At the same time, the expression of HDACs and HATs of Rosa chinensis under high temperature was quantified by real-time PCR. These results gave us more infor about the HDACs and HATs in rose germplasm, and provide a new idea for the research on high temperature resistance of rose. some problems:
- Regarding the abbreviation of HDAC, there are many mistakes in the article. Such as: line 26, line 66, line 480 and line 513, please check again.
- In 3.1, the HDACs and HATs of 6 plant materials were described, however, only three plants were selected for the phylogenetic tree construction in 3.4. What is the basis for the selection?
- In 3.4.2, the description of the text was not consistent with figure 2b, and the same problem exists in 3.4.3.
- The subtitle serial number is wrong. From line 290, the subtitle of the third section is wrong.
- In 3.4.3, how did the author come to the conclusion about "RcHDT1 was more closely related to FvHDT3-1, and RcHDT2 was more closely related to FvHDT3-2 than thaliana and O. sativa." FvHDT3-1 and FvHDT3-2 were not involved in the corresponding picture.
- In 3.5, regarding the sampling of flowers and buds, the variety OB was very different from Rw and Ry. Different flower organs and buds of different periods of OB were taken as samples, while Rw and Ry only took buds as samples and did not collect flowers. However, there are several places in the article that compared the flowers of the three varieties together, such as line 328, line 356, line 374 and line 380). At the same time, which period of OB buds were selected and compared with Rw and Ry.
- In line 442, how did you come to the conclusion that HDA6 was related to root development? references?
- Rosa chinensis 'white pearl in red dragon's mouth ', an ancient Chinese rose variety, the suggestion is revised to Rosa chinensis 'Chilong Hanzhu' according to ICNCP.
- 'three rose varieties' in line 82, include two cultivars and one species, the suggestion is revised to 'three rosa germplasm'.
Reviewer 2 Report
The manuscript examined the role of histone acetylation in heat stress response in rose plants. The series of the experiments were conducted clearly to support the results. However, authors should spend time for checking minor spells in the manuscript. For example: Line 44, please add "." between [7,8] and In; or line 153: please check "ACTING" or "ACTIN".
After plagiarism checking, I found that authors used full sentences from previous published papers, such as lines 59, 60-61; 243-247; 381-382; 463-464; 465; .... Please check and rewrite them with your ideas.
Reviewer 3 Report
A manuscript under review represents the results of a broad genome-wide survey for genes of two families in genomes of Rosa spp. and studies on expression pattern of these genes under heat stress. Authors have applied the most contemporary methods to reach their goal.
In my opinion, there are two major concerns about this paper.
1. When discussing expression levels, authors do not provide any results of statistical treatment. It looks as if they judge about differences only qualitatively, in terms like 'slightly higher' etc. However, all differences, if any, need to be checked for their statictical significance. This is a serious flaw of this paper which needs to be corrected.
2. The whole text of a manuscript needs a serious elaboration considering its language and style. I did not manage to understand some of sentences. In addition to its style, the whole concept of such work (even if written in perfect English) is not so easy to get into, as it contains lots of designations of genes and proteins, abbreviations of cultivars etc. I think authors should avoid overcomplication of their text providing some (or most) quantitative data as tables and/or figures. Unfortunately, at the moment the biggest part of the Results section is a retelling of data already available in figures/tables. I strongly recommend authors to reduce their Results significantly to avoid doubling and help readers.
Additional suggestions and corrections are available in a manuscript file (see attached).
After considering reviewer's comments, it represents the results of a big work. However, these results need to be presented in a better quality.

Round 2
Reviewer 3 Report
In its updated form, this manuscript has been significantly improved. I am glad that numerous abbreviations were removed from the main text making it shorter and easier to read. Some minor issues can be always found, but these will be probably removed in the course of editorial work. For example, 'spp.' should not be italicized (Rosa spp., not Rosa spp.). I recommend the revised manuscript for publication in Genes.